# The Characterization of Titanium Particles Released from Bone-Level Titanium Dental Implants: Effect of the Size of Particles on the Ion Release and Cytotoxicity Behaviour

**DOI:** 10.3390/ma15103636

**Published:** 2022-05-19

**Authors:** Juan Antonio Callejas, Aritza Brizuela, Blanca Ríos-Carrasco, Javier Gil

**Affiliations:** 1Bioengineering Institute of Technology, International University of Catalonia, Josep Trueta s/n, Sant Cugat del Vallés, 08250 Barcelona, Spain; drcallejas@gmail.com; 2Faculty of Dentistry, International University of Catalonia, Josep Trueta s/n, Sant Cugat del Vallés, 08250 Barcelona, Spain; 3Facultad de Odontología, Universidad Europea Miguel de Cervantes, C. del Padre Julio Chevalier 2, 47012 Valladolid, Spain; aritzabrizuela@hotmail.com; 4Periodontics, Facultad de Odontología, Universidad de Sevilla, Avicena s/n, 41004 Sevilla, Spain

**Keywords:** titanium debris, bone level implants, friction forces, chemical degradation, cytotoxicity

## Abstract

Many studies are being carried out on the particles released during the implantoplasty process in the machining of dental implants to remove bacterial biofilms. However, there are no studies on the release of particles produced by the insertion of bone-level dental implants due to the high compressive frictional loads between the rough titanium implant and the bone tissue. This paper aims to characterize the released particles and determine the release of titanium ions into the physiological environment and their cytocompatibility. For this purpose, 90 dental implants with a neck diameter of 4 mm and a torque of 22 Ncm were placed in 7 fresh cow ribs. The placement was carried out according to the established protocols. The implants had a roughness Ra of 1.92 μm. The arrangement of the particles in the bone tissue was studied by micro-CT, and no particle clusters were observed. The different granulometries of 5, 15, and 30 μm were obtained; the specific surface area was determined by laser diffraction; the topography was determined by scanning electron microcopy; and the particles were chemically analysed by X-ray energy microanalysis. The residual stresses of the particles were obtained by X-ray diffraction using the Bragg-Bentano configuration. The release of titanium ions to the physiological medium was performed using ICP-MS at 1, 3, 7, 14, and 21 days. The cytocompatibility of the particles with HFF-1 fibroblast and SAOS-2 osteoblast cultures was characterized. The results showed that the lowest specific surface area (0.2109 m^2^/g) corresponds to the particles larger than 30 μm being higher than 0.4969 and 0.4802 m^2^/g of those that are 5 and 15 μm, respectively, observing in all cases that the particles have irregular morphologies without contamination of the drills used in the surgery. The highest residual stresses were found for the small particles, −395 MPa for the 5 μm particles, and −369 for the 15 μm particles, and the lowest residual stresses were found for the 30 μm particles with values of −267 MPa. In all cases, the residual stresses were compressive. The lowest ion release was for the 30 μm samples, as they have the lowest specific surface area. Cytocompatibility studies showed that the particles are cytocompatible, but it is the smallest ones that are lower and very close to the 70% survival limit in both fibroblasts and osteoblasts.

## 1. Introduction

Titanium dental implants are proving to be a good solution for tooth replacement [1,2,3]. This metal is the most widely used compared to other alloys or ceramics such as zirconia [3,4]. This technology has been developed little by little to improve the osseointegration levels of implants [4,5,6], avoid the adhesion of bacterial plaque [7,8], improve aesthetics [9,10], and facilitate dental surgery [11]. Dental implants with an external connection show a greater facility for bacterial leaching than dental implants with an internal connection. This fact has also led to the idea that dental implants at the bone level also could reduce this bacterial colonization compared to those implanted above the bone level. However, this is a controversial fact that needs more research to be confirmed [7,8,12,13,14].

Peri-implantitis is a disease affecting around 30% of dental implants inserted after 10 years of their placement. This is caused by bacterial colonization and characterized by inflammatory changes in the peri-implant mucosa and progressive bone loss around an osseointegrated dental implant [15]. This biological problem must be treated because if it is not resolved, the implant and prosthetic will fail. Metal particles can be released into the peri-implant tissues in different ways, including implant insertion, corrosion, friction, or passivation methods of the dental implant, such as implantoplasty (IP). Although Ti particles seem to induce the expression of pro-inflammatory cytokines and decrease the viability of osteogenic cells [16,17], the immunological characteristics (inflammatory and osteogenic response) of metal particles released during IP are still unknown.

The placement of dental implants at the bone level requires high frictional forces at the neck of the dental implant with the cortical bone, which gives a state of compression that allows the mechanical fixation of the implant [5,18]. These frictional forces between the cortical bone and the rough surface of the dental implant cause particles to be released during placement [16]. Their size varies according to the roughness of the dental implant. The number of particles depends on the force exerted during placement. The manufacturers of bone-level implants have very detailed protocols for their placement. In our study, we have faithfully followed the protocol, but it is well known that dentists may sometimes use higher stresses to ensure mechanical stability of the bone to the dental implant, which results in a higher number of particles being released.

The relationship between peri-implant diseases and bacterial biofilms has been demonstrated in different research [15,16,17,18,19]. According to the classification of periodontal and peri-implant diseases and factors, there are several conditions that require further investigations to show their effect on the development of peri-implant diseases [20]. Among these factors, particles of titanium and other metals or alloys have been suggested as a risk factor for bone loss and inflammation of the peri-implant mucosa [17,21,22]. Most implants are made of different grades of commercially pure titanium or alloys with other metals, and periimplantitis has been associated with a greater accumulation of Ti in peri-implant tissues in comparison with healthy implants [23,24]. Additionally, it has been reported that Ti debris can promote DNA damage in oral epithelial cells by activating the molecular markers CHK2 and BRCA1 [25,26].

These particles have been very poorly characterized from the point of view of their physico-chemical characterisation. Some studies have been carried out on particles from implantoplasty, but there are practically no studies on particles from bone-level implants, which are generally placed with significant friction forces. This contribution aims to shed light on the characterisation of these particles released during implant placement and their cytotoxic capacity in neighbouring tissues.

Experimental studies [27,28,29,30,31] have demonstrated the importance of the size of the released particles on biological behaviour. It has been studied that particles with small sizes lower than 5–15 μm produce detrimental effects on biological behaviour than particles of the same chemical composition with bigger sizes. This fact is due to the increase in the specific surface and more surface in contact with the tissues with the same mass [32,33]. Hence, the importance of size and surface area does not override the importance of particle composition. Studies have shown that Ti6Al4V dental implants [15] have higher levels of cytotoxicity in detached particles of the same size. Schwarze et al. [26,27] showed that metal particles can induce problems of apoptosis depending on the particle size. The surface reactivity and the chemical composition play a key role in the pro-inflammatory potential of the particles [31,34]. In Orthopedics and Traumatology, the residues released by metal prostheses and how they affect the surrounding tissues have been studied in depth. These particles are released by the continuous wear and tear between the articular surfaces [35].

In this study, we want to determine the effect of size on the release of titanium ions into the physiological environment, i.e., their chemical degradation and how it affects the levels of cytocompatibility with fibroblastic and osteoblastic cells.

## 2. Materials and Methods

### 2.1. Dental Implants and the Collection of Detached Particles

Ninety bone-level dental implants made of titanium grade 4 were inserted in fresh cow ribs by the same investigator (J.A.C.) according to the surgical protocol described by the company [36]. The design of the bone level dental implant used (Vega plus, Klockner, Escaldes Engordany, Andorra) can be observed in Figure 1. These dental implants were sandblasted with alumina from 300 to 600 μm and after were passivated by citric acid at 20% for 10 s. The roughness obtained was R_a_ of 1.92 and an R_z_ of 1.87 μm and a diameter of 4.0 mm in the neck and 9 mm in length with a conical shape.

A total of 90 dental implants were implanted in 7 fresh cow ribs (15 for each rib) with a distance between implants of 9 mm. The cow ribs present a length range from 210 mm to 230 mm. The bone density was homogenous in all the segments and ribs. The surgeries were realised by the same researcher (J.A.C.). The process of implantation is shown in Figure 2 using a GENTLEsilence LUX 8000B turbine (KaVo Dental GmbH, Biberach an der Riβ, Germany) under water irrigation at 22 °C [37].

When the 90 dental implants were placed in the bone, shorts were made to separate each of the placed dental implants. All samples were analysed by a high–resolution micro–CT scanner (Skyscan 1272CMOS, Bruker, Billerica, MA, USA) (Figure 3). The metal particles detached from the implant could be observed in the bone tissue. Due to the few particles and their small size, the scanning speeds were very low to obtain a higher resolution.

After micro-CT analysis, the samples were placed in an oven at 920 °C for 5 h to remove all organic content from the bone tissue. Subsequently, the mineral content (apatite) and the detached titanium particles were obtained and separated by flotation [36].

### 2.2. Specific Surface Area

The specific surface area understood as the contact surface of the particle with the physiological medium was determined by an ASAP 2020 equipment (Micromeritics, Norcross, GA, USA) in vacuum conditions below 10 µmHg and using nitrogen as adsorbate. The particles were degassed at 100 °C. The specific surface area was determined by mathematical calculations according to BET (Brunauer–Emmett–Teller) theory [24].

### 2.3. Granulometry

The size of the detached particles was determined using the Mastersizer 3000 (Malvern Panalytical, Malvern, UK). This unit uses the laser diffraction technique to measure particle size by measuring the intensity of scattered light as a laser beam passes through the sample of particles.

The test is performed in alcoholic medium (ethanol) used as liquid scattering medium. The equipment allows the analysis of particles between 9 nm and 3.45 mm. To avoid agglomeration of particles during the particle size test, two types of agitation are used: mechanical agitation using a 2500 rpm shaker and ultrasonic agitation at 50% sonication to ensure adequate measurement of all isolated particles.

### 2.4. Scanning Electron Microscopy

Particle morphology was performed using a scanning electron microscope, the Jeol 6400 Scanning Electron Microscopy (JEOL, Tokyo, Japan), with a resolution power of 15 nm and an acceleration voltage of 20 KeV. Gold coating of the surfaces by sputtering was not necessary as the samples were sufficiently conductive. In addition, the microscope was coupled with an energy dispersive X-ray microanalysis system EDS Oxford, Oxford, UK).

### 2.5. Residual Stresses Determination

Residual stresses were obtained from the particles using an X-ray diffraction apparatus incorporating the Bragg-Brentano configuration (D500, Siemens, Munchen, Germany). The stresses exerted on the metal during machining cause deformation and therefore variation of the lattice parameters that give a micro-strain since we know the elastic constants are translated into surface tension levels.

The families of planes (213) were used to make the measurements as these planes diffract at 2θ = 139.5°. Furthermore, the elastic constants of titanium in this family of planes are well known EC = (E/1 + υ) (213) = 90.3 (1.4) GPa. Eleven different ψ angles were evaluated: 0° and five positives and five negatives. The distances of the peaks are fitted with the pseudo-Voigt function using the software (WinplotR, Rennes, France); this allows the conversion to the interplanar distances using Bragg’s law. The dψ vs. sen2ψ graphs permitting the determination of the slope of the linear regression (A) were completed with software (Origin, Microcal, Amherst, MA, USA). The residual stress is σ = EC (1/d_0_) A, where d_0_ is the interplanar distance for ψ = 0°.

### 2.6. Ion Release

Titanium ion release analyses were performed on five samples random (n = 5) for each of the sizes according to ISO 10993-12-2009. For this purpose, the ratio of medium to solid particles of 1 mL per 0.2 g of particles used was used in accordance with the standard. In our study, 10 mL of medium was prepared for analysis, which corresponds to 2 g of particles per test.

The liquid medium used for ion release was Hank’s saline solution (Sigma-Aldrich, Co., Life Science, St. Louis, MO, USA). Hank’s solution in contact with the particles was recovered and filtered through a filter with a pore size of 0.22 µm. For analytical study, the solution was acidified with 2% nitric acid (HNO_3_ 69.99%, Suprapur, Merck, Darmstadt, Germany) to avoid precipitation of the metal ions prior to measurement of their concentration by inductively coupled plasma emission mass spectrometry (ICP–MS).

Extractions for analysis were performed at 1, 3, 7, 7, 14, and 21 days following other similar studies [38,39]. The samples were kept at 37 °C in an oven and shaken at 250 rpm, varying the inclination from 0 to 30° in order to avoid settling the metal debris during the test and to ensure continuous exposure of all particles to the medium. The samples were analysed by ICP–MS (Perkin Elmer Elan 6000, Perkin Elmer Inc., Waltham, MA, USA). This method permits quantitative multi-elemental analysis with an accuracy of 1 ppt for titanium.

### 2.7. Preparation of Samples for Cell Cultures

Each sample was independently sterilized with 96% ethanol separately prior to cell culture to determine cytocompatibility. Ethanol immersion of the samples lasted 30 min. Subsequently, the alcohol was removed by three cycles of centrifugation at 7200 rpm for 5 min each cycle. The samples were washed with Dulbecco’s Phosphate-Buffered Saline (DPBS) (Sigma-Aldrich^®^, Sant Louis, MO, USA). Washing was repeated three times. In accordance with ISO 10993-5, cell culture medium was added after each centrifugation to maintain the concentration of 0.2 g of particles per mL of medium.

### 2.8. Cytotoxicity Test

The cytotoxicity tests were realized in triplicate (n = 3); the samples were evaluated by indirect exposure determination following the standard ISO 10993. The positive control of the test used the plate seeded with cells, and the negative was the medium without cells.

The samples were always treated under aseptic conditions. The percentage of cells that survive when exposed to the medium that was in contact with the particles was determined for each of the cell lines. This is called the indirect assay and complies with the guidelines specified in ISO 10993-5 “Biological evaluation of medical devices”, part 5 “In vitro cytotoxicity tests”. In this standard, the material is considered to be cytotoxic when the surviving cells are less than 70%. In our study, with particles that are released in bone and soft tissue, studies were carried out with human cell lines: SAOS-2 osteoblastic (ATCC^®^ HTB-85, New York, NY, USA) and HFF-1 fibroblastic cells (ATCC^®^ SCRC-1041, New York, NY, USA). THP-1 macrophage line (DSMZ, ACC 16) was used to evaluate the inflammatory reaction. Cell lines were cryopreserved at −180 °C in dimethyl sulfoxide for storage and assayed every two months for the absence of mycoplasma.

Cultures were carried out in a 5% CO_2_ incubator under humid conditions. For cultures with SAOS-2 cells; Dulbecco’s Modified Eagle Medium (DMEM; Thermo Fisher Scientific, Waltham, MA, USA) with an addition of 10% Foetal Bovine Serum (FBS; Thermo Fisher Scientific, Waltham, MA, USA), 1% L-glutamine (Thermo Fisher Scientific, Waltham, MA, USA), and 1% penicillin/streptomycin (Thermo Fisher Scientific, Waltham, MA, USA) were used. For HFF-1 fibroblast cells (RPMI), 1640 medium (Sigma-Aldrich^®^, Sant Louis, MO, USA) with the addition of 10% foetal bovine serum (FBS) (Sigma-Aldrich^®^, Sant Louis, MO, USA) and 1% penicillin–streptomycin (Fisher Scientific, Hampton, USA) was used for THP-1 culture. The liquid was kept at −4 °C.

The extracts were analysed according to Section 8.2 of ISO 10993-5. The material was placed in liquid medium at a ratio of 1 mL per 0.2 g of sample and kept at 37 °C for 72 h. The cells were seeded prior to contact with the sample extracts at a density from 2 to 104 cells/mL for 24 h.

Cells were incubated for 24 h with undiluted and 1/2, 1/10, 1/100 and 1/1000 diluted extract using complete medium for dilutions. Cells were analysed for adhesion before and after contact with the extracts. When the test was finished, cells were lysed with Mammalian Protein Extraction Reagent (mPER), and cell viability was evaluated as lactate dehydrogenase enzyme activity (LDH; Roche Applied Science, Penzberg, Germany). Viability was determined following the manufacturer’s recommendations, measuring absorbance at 492 nm.

### 2.9. Statistical Analysis

The results were recorded by a Microsoft Excel spreadsheet (Microsoft^®^, Redmond, Washington, DC, USA). After, the measurements were processed with the Stata 14 package (StataCorp^®^, College Station, San Antonio, TX, USA). Statistical values such as averages and standard deviations were calculated, except for the granulometry test, where the mode and percentiles were used.

## 3. Results and Discussion

Observation of the micro-CT images showed particle dispersion with no agglomeration. The areas where the vast majority of particles were seen were in the cortical zone of the bone. Very few particles appeared in the cancellous tissue, and in these cases, they came from the cortical zone from which they had been entrained.

In Figure 4, we can see that the particle size distributions of 5, 15, and 30 μm obey a Gaussian curve. The averages of the equivalent diameters are shown in Table 1.

The specific surface values are shown in Table 2. These indicate that small samples have the highest specific surface area and are therefore the most reactive with the surrounding physiological environment. The differences between the specific surface areas of the Ti-5 µm and Ti-15 µm samples are not statistically significant. On the other hand, the Ti-30 µm sample has a significantly lower specific surface area than the other samples. That is, for 30-micrometre particles, it has less reactivity per unit weight than the smaller ones.

The particle morphology of the three samples studied (Figure 5) is irregular as a consequence of the use of synthesis processes by mechanical milling and which very accurately reflects the particles found in the tests carried out on the veal rib.

Figure 6 shows an energy-dispersive X-ray diffractogram. More than 20 were made for each of the samples, and only the peak typical of commercially pure titanium was visible. Therefore, we can say that the particles did not show any contaminants in their structure.

In the microtomographic analyses in which the titanium particles released in the placement of the rib were detected, the presence of residues of the AISI 304 martensitic stainless steel of the drills was not observed. No residues were confirmed by the microanalysis of the X-ray (Figure 6).

Martensitic stainless steel is magnetic, and we should have picked it up. However, all the metal residues were non-magnetic and therefore corresponded to titanium. The absence of martensitic steel in the tissues is very important due to the corrosion of this steel. These residues are considered toxic [38,40].

The values for the residual stresses are summarized in Table 3. As expected, smaller particles absorb more energy (compressive residual stress) due to the fact they have less material mass, and therefore, the residual stresses are higher. There are no statistically significant differences between T-5 µm and Ti-15 µm, but there are significant differences (*p* < 0.001, *t*-Student) between these sizes and Ti-30 µm, which are larger.

This higher energy makes the material more reactive, and therefore, it will have a worse corrosion resistance [39,41].

Figure 7 shows the three titanium ion release curves in the liquid medium (Hanks’ solution) as a function of incubation time. The qualitative comparative analysis of the three curves shows a similar behaviour of the three powder samples studied, with an initial rapid growth of titanium ions released (3 days) that continues with slower growth and reaches quasi-stabilization (from 3 days to 21 days of immersion).

Quantitative comparative analysis of the three curves showed differences between the samples in the number of ions released as a function of incubation time. The sample Ti-30 µm showed significantly lower ion release values than the other two samples evaluated, Ti-5 µm and Ti-15 µm. The lower release of Ti ions shown by the Ti-30 µm sample could be explained by its larger particle size, which would lead to a smaller specific surface and, consequently, to a smaller area exposed to the medium, as can be seen in the results shown in Table 2.

The Ti-5 µm and Ti-15 µm samples showed practically the same titanium ion release values as a consequence of their similar particle size, as can be seen in the results in Table 2. However, there is a tendency for Ti-15 µm to show a higher release, although the results are not statistically significant.

These results are also in agreement with the residual stress results. Samples with higher residual stresses are more unstable and therefore facilitate their chemical degradation by increasing the release of ions into the medium [39,41,42,43,44].

The results do not show statistically significant differences with *p* < 0.05, which is the normal value in statistical analyses, but they show differences with *p* < 0.1 and are marked with the symbol *. The results between Ti-5 µm and Ti-15 µm do not present statistically significant differences, and these samples in comparison with Ti-30 µm present the following *p*-values: for 5 days, *p* = 0.082; for 7 days, *p* = 0.067; for 15 days, *p* = 0.086; and for 25 days, *p* = 0.067).

The results obtained from the cytotoxicity test expressed as a percentage of cell survival are shown below (Figure 8). As a rule, 70% is established as the lower limit of cell survival for a material not to be considered cytotoxic.

The cells were adherent to the substrate plate and showed the expected morphology, both before and after incubation with the extracts.

## 4. Discussion

Micro-CT studies have shown that the cortical tissue is the priority site for the released particles. Their morphology is irregular, and they are of three different sizes: 5, 15, and 30 μm. This shape is due to the fracture of the peaks of the rough titanium surface due to frictional forces. Therefore, there may be an influence on the sizes due to the different roughnesses of the dental implants. At present, most dental implants have a roughness between 1.5 and 2.2 μm, so the differences in the sizes of the roughened particles will be very similar. It can be seen that the granulometry curves have a rather small width, which means that there is a good reproducibility of the sizes. No different size families or traces of possible impurities in the powder that would show an anomalous peak in the size distribution or granulometry are observed.

In some particle works, it is observed that the particle shapes are spherical, a fact that does not follow the reality in which the samples observed in the tissues are irregular. This irregular shape is the result of the fracture of the roughness peaks when the dental implant is placed in friction against the cortical bone. When work is carried out with spherical particles, they do not faithfully follow reality, and this is due to the fact that the particles are not obtained mechanically: they are obtained by atomization. In this technique, the particles are obtained by solidifying a liquid spray of titanium and, due to the surface energy, the liquid tries to form spherical geometries in order to reduce the energy. The sphere is the largest possible volume with the smallest possible surface area, and this produces minimum interface energy [30].

As the particle size results show, no large differences in particle size are observed between the Ti-5 µm and Ti-15 µm samples, but large differences in particle size are observed with the 30 µm samples. What is apparent from the electron microscopy images is the regularity of the particle shapes and the homogeneity of the particle sizes.

It would be a mistake to compare the particles produced by the frictional forces in the placement of bone-level dental implants with those produced by the implantoplasty technique. In the latter case, the particles are generated by mechanization, making the process of obtaining them different and much more aggressive.

As we have seen, the residual stresses are always compressive and are of greater value in the smaller particles. This fact is due to the fact that the fracture work to free the particle from the substrate is the same, but in the smaller particles, a greater amount of stress is stored as the volume is smaller. This fact will favour electrochemical corrosion, as these particles are the most unstable, and corrosion will reduce the residual stress. Therefore, this factor should be studied, as corrosion products can be cytotoxic.

The lower release of Ti ions shown by the Ti-30 µm sample could be explained by its larger particle size, which would lead to a smaller specific surface and, consequently, to a smaller area exposed to the medium, as can be seen in the results shown in Table 1.

The Ti-5 µm and Ti-15 µm samples have shown practically the same titanium ion release values as a consequence of their similar particle size, as can be seen in the results in Table 1. However, there is a tendency for Ti-5 µm to show a higher release, although the results are not statistically significant. These results are also in agreement with the residual stress results. Samples with higher residual stresses are more unstable and will therefore facilitate their chemical degradation by increasing the release of ions into the medium [42,45].

The samples evaluated in the cytotoxicity test showed values slightly above 70% cell survival. In some cases, they showed lower values, although due to the variability of the test, there were no significant differences. It can be seen from the results of the cytotoxicity figures that it is the smaller particles that show lower cytocompatibility values, bordering on cytotoxicity in terms of average values [39,46,47]. However, there are small particles that can be said to be cytotoxic, although strictly speaking, this cannot be affirmed since, in general, the mean values are higher than 70%.

In the study by Vara et al. [16], it was determined that particles released from the Ti6Al4V alloy show a higher degree of cytotoxicity than those presented in this study of commercially pure titanium. It is also important to note that the release of vanadium ions is very significant and may be affecting the low level of cytocompatibility of the Ti6Al4V alloy. In our case, the titanium release for all sizes studied is low and does not affect the cytocompatibility.

This study has the limitation that in vivo testing is not possible. It is for this reason that we attempted to simulate the process of implant insertion in fresh cow rib bone using the insertion protocols used in the placement of dental implants in patients. The implantation was performed by an experienced surgeon trying to always perform the surgery in the same manner and following the established protocols. The samples were characterized according to international standards for each type of test. However, one limitation is that sometimes surgeons do not follow the protocols set by the commercial companies exactly, and this can differentiate the results obtained.

Future lines of research based on this work include the study of inflammatory reactions and the response to macrophages, which may shed light on the effect of the particles on peri-implantitis disease. It would also be very appropriate to be able to study the corrosion of these particles in a physiological medium, as the corrosion products could be cytotoxic. It would also be important to compare the particles released in the processes of implantoplasty with those released in the placement of dental implants at the bone level. We believe that this article contributes to the knowledge of the characteristics of the particles, but it is now necessary to determine their biological and microbiological influence and their effects on the generation of peri-implant disease.

## 5. Conclusions

Particles released at the insertion of bone-level dental implants were characterized. It was determined by micro-CT that the released particles are located in the cortical bone in isolation and are of 5, 15, and 30 μm sizes. The particles are irregular, with the 30 μm particles having the smallest specific surface area, which leads to a lower release of titanium ions into the physiological environment at all times studied. The particles of smaller sizes 5 and 15 μm present the highest compressive residual stress, which indicates that they have a greater tendency to corrosion. In all cases, the particles are cytocompatible, although the cell survival values in both human fibroblasts and osteoblasts are smaller in the smaller ones.

## Figures and Tables

**Figure 1 materials-15-03636-f001:**
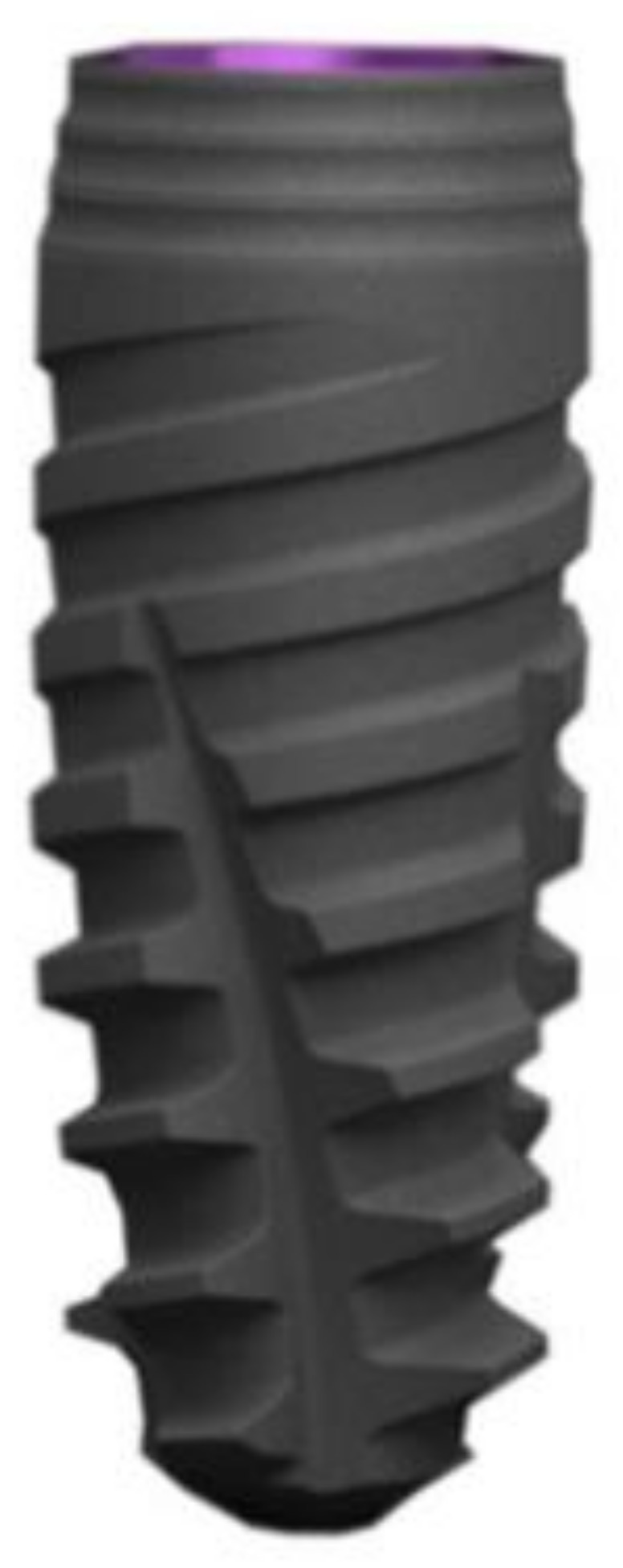
Commercially pure titanium grade 4 bone level dental implant (Vega plus by Klockner Dental Implant System).

**Figure 2 materials-15-03636-f002:**
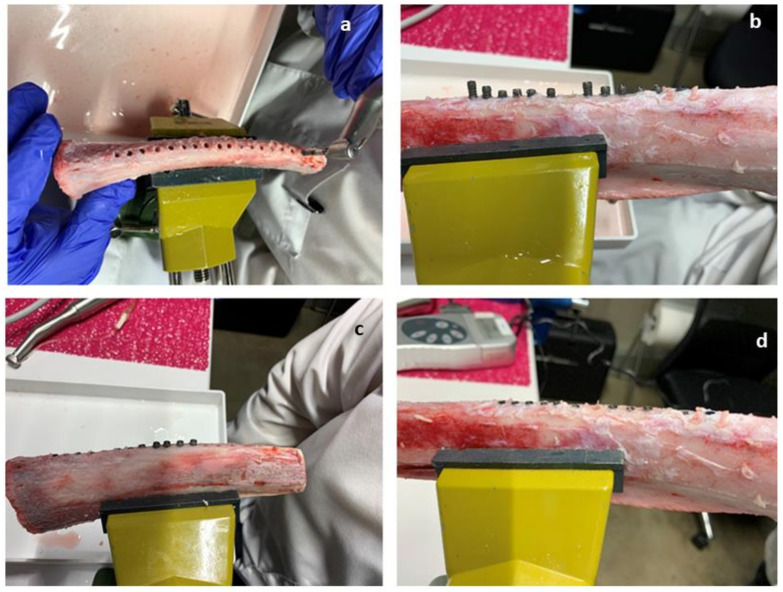
Bone-level dental implant inserted in the fresh cow ribs. (**a**) Drilling of the bone to 3.5 mm in diameter. (**b**) Dental implant insertion process with a torque of 22 Ncm. (**c**) Fixation of the implant at bone level at 22 Ncm. (**d**) Final placement at bone level.

**Figure 3 materials-15-03636-f003:**
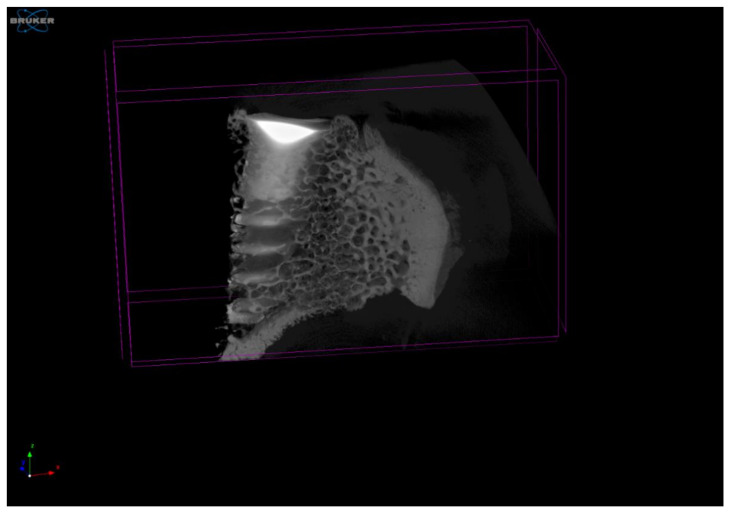
Image of the dental implant in the rib of a cow observed by high-resolution micro-CT. Micro-CT was used to view the dental implant surrounded by cortical and cancellous bone tissue. It was determined that the detached particles were in the cortical tissue, and no accumulations of particles were observed, but the particles were isolated.

**Figure 4 materials-15-03636-f004:**
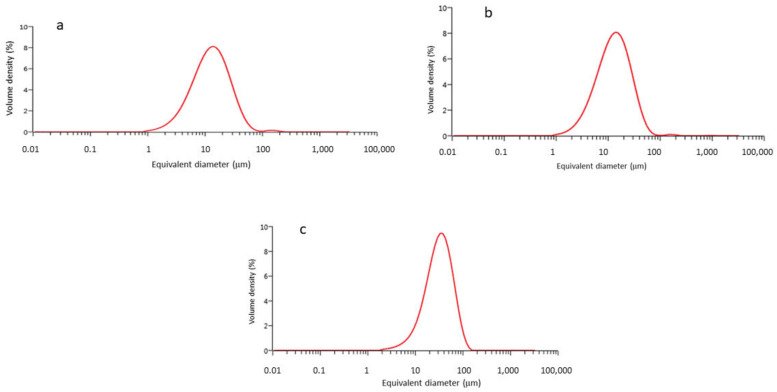
(**a**) Granulometry for Ti-5 μm, (**b**) Ti-15 μm, and (**c**) Ti-30 μm.

**Figure 5 materials-15-03636-f005:**
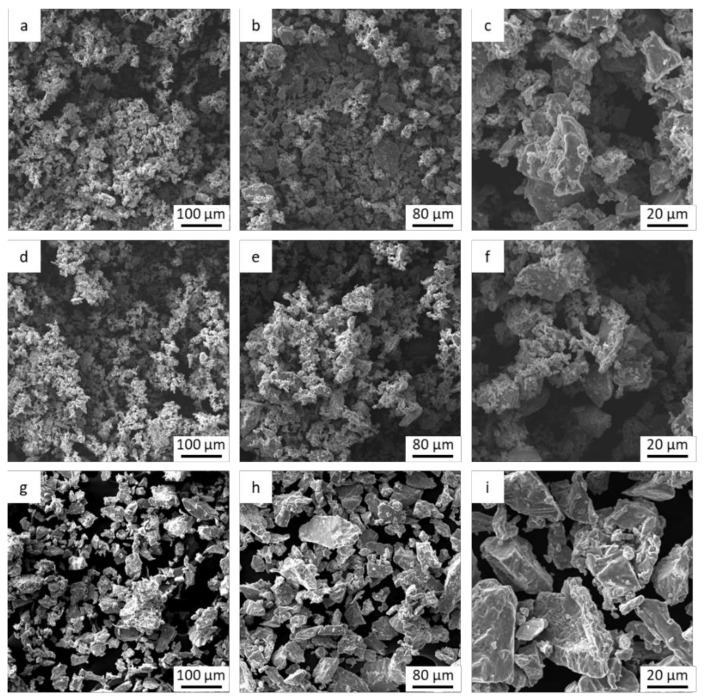
Particles observed by scanning electron microscopy for Ti-5 µm (**a**–**c**), Ti-15 µm (**d**–**f**), and Ti-30 µm (**g**–**i**) samples.

**Figure 6 materials-15-03636-f006:**
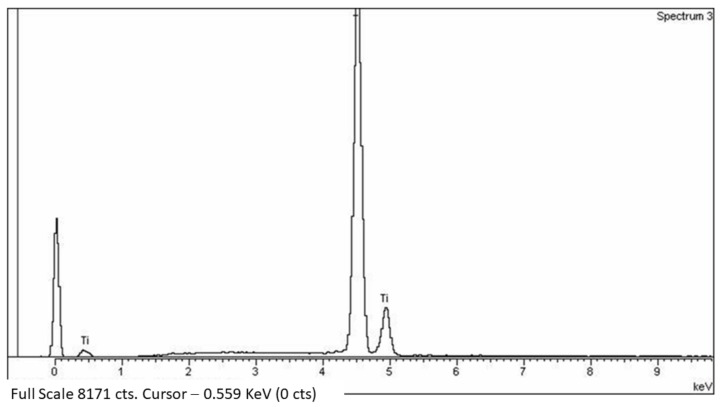
Microanalysis of X-ray obtained in a Ti-5 µm sample.

**Figure 7 materials-15-03636-f007:**
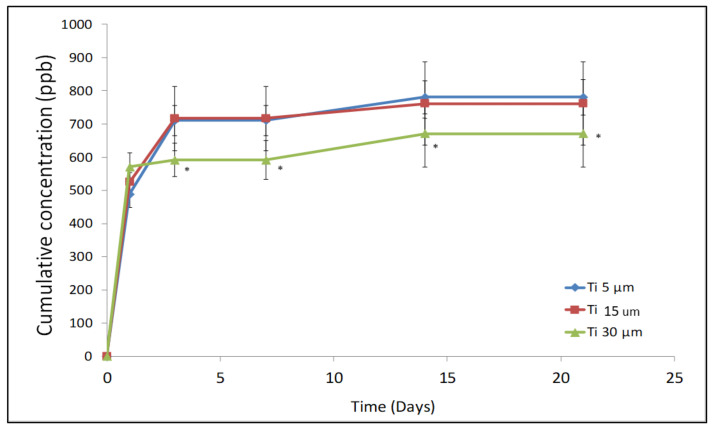
Ion release for the different sizes of c.p.Ti particles at different testing times (* menas statitstical differences significance with *p* < 0.1).

**Figure 8 materials-15-03636-f008:**
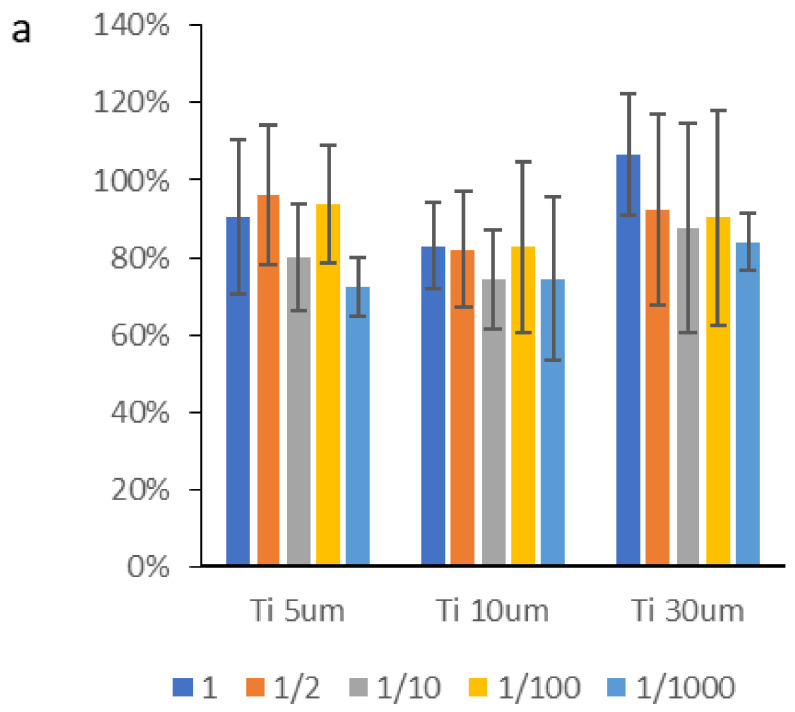
Cellular viability in cytotoxicity tests. (**a**) Osteoblasts’ SAOS-2 cells. (**b**) Fibroblasts’ HFF-1. The results do not present statistically significant differences, *p* < 0.001 (Osteoblasts’ SAOS-2 *p* = 0.011 and fibroblasts’ HFF-1 *p* = 0.006).

**Table 1 materials-15-03636-t001:** Average of equivalent diameters for each size studied.

Samples	Average Equivalent Diameter (μm)
Ti-5 μm	7.3
Ti-15 μm	14.9
Ti-30 μm	32.3

**Table 2 materials-15-03636-t002:** Specific surfaces for the different size of samples.

Samples	Specific Surface (m^2^/g)
Ti-5 μm	0.4969 ± 0.0037
Ti-15 μm	0.4802 ± 0.0042
Ti-30 μm	0.2109 ± 0.0009 *

* means statistically significant differences, *p* < 0.001.

**Table 3 materials-15-03636-t003:** Residual stresses for the different sizes of samples.

Samples	σ (MPa)
Ti-5 μm	−395 ± 27
Ti-15 μm	−369 ± 32
Ti-30 μm	−267 ± 21 *

* means statistically significant differences, *p* < 0.001.

## Data Availability

The datasets generated during and/or analysed during the current study are available from the corresponding author on reasonable request.

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
