# Peer review of "The Characterization of Titanium Particles Released from Bone-Level Titanium Dental Implants: Effect of the Size of Particles on the Ion Release and Cytotoxicity Behaviour"

_materials, 2022, doi:10.3390/ma15103636_

Round 1
Reviewer 1 Report
Introduction section
- Please provide the reference as evidence for the sentence of “The placement of dental implants at bone level requires high frictional forces at the neck of the dental implant with the cortical bone which gives a state of compression that allows the mechanical fixation of the implant.”
- The sentence of “It has been observed that have shown that” need to correct the grammar error.
- Please provide the reference as evidence for the sentence of “particles with small sizes lower than 10-15m produce detrimental effects on biological behavior than particles of the same chemical composition with bigger sizes.”
Materials and methods section
- Please provide the information related to the size of the dental implant, are they the same size?
- What is the purpose of scanning the sample byMicro-CT?, because particles are difficult to calibrate and quantify from micro-CT images.
- How to remove all organic substances and leave only mineral content (apatite) through this procedure (only in an oven at 920°C for 5 h)? Are there other procedures?
- A big problem in the section of materials and methods is that many analyses did not clearly indicate the number of samples. Please provide them in each analysis.
- Another big problem in this paper is that only averages and standard deviations are not statistical analysis. Statistical analysis must be determined whether to parametric test or nonparametric test, and after statistical analysis to determine whether it has reached statistical significance! It is recommended that the results must have actual statistical analysis, specifically for Table 3, Table 4, Figure 9, Figure 10.
Results and Discussion section
- In my opinion, without statistical analysis (p value). it looks like that no significance different between the results of Ti-5μm and Ti-15μm in Table 3, Table 4, Figure 9.
- In Figure 10, the results must provide an analysis of whether they are statistically significant (p value), otherwise they are not statistically significant from the picture.
Author Response
REVIEWER 1
Dear Reviewer,
Thanks for taking the time to review our manuscript and suggest to us to improve our work by providing a lot more detail. We have done so, and we are now submitting a manuscript that not only addresses the points the you specifically raised but also many others that we have considered in order to deliver what we think is a much improved version of our work. This version includes more paragraphs, English grammar revisions in all main sections, new references. Thanks a lot. We are looking forward to your comments.
Sincerely,
Francisco-Javier Gil Mur
Introduction section
- Please provide the reference as evidence for the sentence of “The placement of dental implants at bone level requires high frictional forces at the neck of the dental implant with the cortical bone which gives a state of compression that allows the mechanical fixation of the implant.”
Two new references were introduced in the text.
- The sentence of “It has been observed that have shown that” need to correct the grammar error.
The sentence has been corrected.
- Please provide the reference as evidence for the sentence of “particles with small sizes lower than 10-15mm produce detrimental effects on biological behavior than particles of the same chemical composition with bigger sizes.”
Three new references have been introduced in the text.
Materials and methods section.
- Please provide the information related to the size of the dental implant, are they the same size?
Yes, the information has been provided in the text according to the reviewer.
- What is the purpose of scanning the sample byMicro-CT?, because particles are difficult to calibrate and quantify from micro-CT images.
The reviewer is right, thank you very much. A text has been incorporated in the legend of figure in order to clarify the purpose.
- How to remove all organic substances and leave only mineral content (apatite) through this procedure (only in an oven at 920°C for 5 h)? Are there other procedures?
Yes, this procedure is used to eliminate all organic compounds from the bones. At this temperature the organic residues evaporate and only the apatite and implant metals remain. A reference to this methodology has been added.
- A big problem in the section of materials and methods is that many analyses did not clearly indicate the number of samples. Please provide them in each analysis.
The number of samples has been clarified in Materials and Methods.
- Another big problem in this paper is that only averages and standard deviations are not statistical analysis. Statistical analysis must be determined whether to parametric test or nonparametric test, and after statistical analysis to determine whether it has reached statistical significance! It is recommended that the results must have actual statistical analysis, specifically for Table 3, Table 4, Figure 9, Figure 10.
In my opinion, without statistical analysis (p value). it looks like that no significance different between the results of Ti-5μm and Ti-15μm in Table 3, Table 4, Figure 9.
The reviewer is absolutely right, and the authors have re-performed the statistical studies. Statistically significant differences have been observed, as indicated by the reviewer, between the values of the smaller particles with respect to the larger ones in the specific surface area and residual stresses with a p<0.001. However, the values for ion releases show differences between the small particles with respect to the 30 micrometer size with a higher p-value p<0.1.In this case, the p-values have been introduced in the text. The cytocompatibility results, as indicated by the reviewer, show no statistically significant differences, although a trend is estimated. Thank you very much for your comment which helps significantly in the interpretation of the results.
Results and Discussion section
- In Figure 10, the results must provide an analysis of whether they are statistically significant (p value), otherwise they are not statistically significant from the picture.
Yes, the results do not present atatistically significant differences in the cytocompatibility in both humans cells. This comment has been introduced in the text
Reviewer 2 Report
Dear authors,
According to my peer review, your manuscript - Characterization of titanium particles released from bone-level titanium dental implants. Effect of the size particles on the ion release and cytotoxicity behaviour (materials-1715767) , falls within the scope of Materials and highlights a widely studied subject in Oral Implantology during the last years. The presence of titanium (Ti) particles around dental implants has been reported in the literature for decades. Although this subject is not new, authors provide an accurate, inclusive, and well-presented research work However, in my opinion a major review is required according to the following aspects:
- Abstract text should be structured. In addition, it does not clarify research aims or summarise statistical information.
- In my opinion there is limited evidence to propose in the Introduction that: " implants placed above bone level is also a lower incidence of bacterial infiltration which reduces the occurrence of peri-implantitis. Authors should review this content
-
A moderate English review is suggested.
-
Introduction can be improved, highlighting the research questions proposed for this study.
-
The rationale for the sample size presented in Materials and Methods should be justified.
-
In Materials and Methods it is referred that "When the 90 dental implants were placed in the bone, shorts were made to separate each of the placed dental implants" How was this preformed in order to obtain equal parts of surrounded bone to the subsequent evaluations?
- More information should be given regarding the surface treatment of the tested implants (Vega plus, Klockner, Escaldes Engordany, Andorra)
- Why titanium ion release was only evaluated on five samples? Were they randomly chosen? What was the criteria for choosing this five samples?
-
Procedures from 2.2 to 2.8 were preformed by the same operators? Who preformed these activities? This information is missing.
- A poor description of statistical analysis was found in your manuscript.
-
Why did authors annexed Discussion to Results? I do not agree with this structure. Discussion should be a independent section.
-
Ninety bone leve implants were placed in fresh cow ribs. How many were placed for rib? What was the distance between them? Did these zones represent the same bone density, promoting the same frictional forces during implant insertion? This latter aspect was not even discussed.
- Which kind of contribution does this study might give to further studies and clinical research?
Author Response
REVIEWER 2
Dear Reviewer,
Thanks for taking the time to review our manuscript and suggest to us to improve our work by providing a lot more detail. We have done so, and we are now submitting a manuscript that not only addresses the points the you specifically raised but also many others that we have considered in order to deliver what we think is a much improved version of our work. This version includes more paragraphs, English grammar revisions in all main sections, new references. Thanks a lot. We are looking forward to your comments.
Sincerely,
Francisco-Javier Gil Mur
Dear authors,
According to my peer review, your manuscript - Characterization of titanium particles released from bone-level titanium dental implants. Effect of the size particles on the ion release and cytotoxicity behaviour (materials-1715767) , falls within the scope of Materials and highlights a widely studied subject in Oral Implantology during the last years. The presence of titanium (Ti) particles around dental implants has been reported in the literature for decades. Although this subject is not new, authors provide an accurate, inclusive, and well-presented research work However, in my opinion a major review is required according to the following aspects:
- Abstract text should be structured. In addition, it does not clarify research aims or summarise statistical information.
The abstract has been improved by providing the objective of the study, more details of the materials and methods as well as the most important results with quantitative data. The statistical study has been performed on all results to determine the statistical significance of the results.
- In my opinion there is limited evidence to propose in the Introduction that: " implants placed above bone level is also a lower incidence of bacterial infiltration which reduces the occurrence of peri-implantitis. Authors should review this content.
The text has been changed
- A moderate English review is suggested.
English has been revised
- Introduction can be improved, highlighting the research questions proposed for this study.
The introduction has been improved with new paragraphs. The state of the art has been analysed and the interest of this paper about the characterization of the titanium debris produced by the frictional forces in the dental implant insertion,
- The rationale for the sample size presented in Materials and Methods should be justified.
The authors have improved the materials and method specially the number of samples, ribs and other details in order to clarify the methodology.
In Materials and Methods it is referred that "When the 90 dental implants were placed in the bone, shorts were made to separate each of the placed dental implants" How was this preformed in order to obtain equal parts of surrounded bone to the subsequent evaluations?
A new details have been incorporated in this part of the Materials and Method in order to calrify the procedure.
- More information should be given regarding the surface treatment of the tested implants (Vega plus, Klockner, Escaldes Engordany, Andorra).
New information about the chacracteristics of the dental implants used have been added.
- Why titanium ion release was only evaluated on five samples? Were they randomly chosen? What was the criteria for choosing this five samples?
Ion release tests follow an international standard ISO 10993-12-2009, that we have followed. Samples must contain at least 150 mg of the sample in order to make dilutions. We have taken for each material this amount in material taken at random for each size. The authors have added more details (random) in the text.
- Procedures from 2.2 to 2.8 were preformed by the same operators? Who preformed these activities? This information is missing.
The researcher JAC performed all surgeries so that the same methodology was applied as indicated in the protocol. The other tests were conducted by him and other members of the research team, so it is not described who conducted each test.
- A poor description of statistical analysis was found in your manuscript.
The reviewer is absolutely right, and the authors have re-performed the statistical studies. Statistically significant differences have been observed between the values of the smaller particles with respect to the larger ones in the specific surface area and residual stresses with a p<0.001. However, the values for ion releases show differences between the small particles with respect to the 30 micrometer size with a higher p-value p<0.1. The p-values have been introduced in the text. The cytocompatibility results, as indicated by the reviewer, show no statistically significant differences, although a trend is estimated. Thank you very much for your comment which helps significantly in the interpretation of the results.
- Why did authors annexed Discussion to Results? I do not agree with this structure. Discussion should be a independent section.
Done
- Ninety bone leve implants were placed in fresh cow ribs. How many were placed for rib? What was the distance between them? Did these zones represent the same bone density, promoting the same frictional forces during implant insertion? This latter aspect was not even discussed.
According to the reviewer we have added more details about the ribs, length, homogeneity of bone density, placement of implants with neck diameter, placement distances between implants to avoid modifications of the bone characteristics among the most remarkable aspects. The placement of the implants was done by the same surgeon following the protocols that are referenced and we think that with 90 implants the possible failures of surgery in any implant can affect very little to the results obtained given the large number of implants used.
- Which kind of contribution does this study might give to further studies and clinical research?
The authors have introduced new paragraphs in the introduction with new references and a new section about the limitations of this contribution and the future studies.
Reviewer 3 Report
Thank you so much for your submission.
I can see the article from authors on ‘Effect of the Nature of the Particles Released from Bone Level Dental Implants: Physicochemical and Biological Characterization’ published recently, is this current paper an extension of the previous paper, if it is please can you mention it in current manuscript?
Introduce the abbreviations first
Ethical approval for animal study, was it taken?
Do not start the sentence with Figure 1 and so on
Clear and zoomed in, well copped photos of implants in cow ribs would be better, in figure 2, the 4 photos need to be highlighted in caption as well
Figure 3- explain and label the figure to explain what it shows
10 µm-no gaps while mentioning this
Extractions for analysis were performed at 1, 3, 7, 7, 14 and 21 days following other similar studies [XXX], please correct XXX?
Figure 8, any clear fill photo available to put in with full margins?
Add in the limitation and future recommendation for your study in clinical practicality based on your results.
Author Response
REVIEWER 3
Dear Reviewer,
Thanks for taking the time to review our manuscript and suggest to us to improve our work by providing a lot more detail. We have done so, and we are now submitting a manuscript that not only addresses the points the you specifically raised but also many others that we have considered in order to deliver what we think is a much improved version of our work. This version includes more paragraphs, English grammar revisions in all main sections, new references. Thanks a lot. We are looking forward to your comments.
Sincerely,
Francisco-Javier Gil Mur
Thank you so much for your submission.
I can see the article from authors on ‘Effect of the Nature of the Particles Released from Bone Level Dental Implants: Physicochemical and Biological Characterization’ published recently, is this current paper an extension of the previous paper, if it is please can you mention it in current manuscript?
In the work cited by the reviewer, the difference in behaviour between commercially pure titanium particles and Ti6Al4V particles was studied. In this work we focus the study on determining the effect of particle size of the same c.p. Ti material on ion release and compatibility. In the discussion of results we have introduced the cite of this paper in order to discuss the difference in toxicity between titanium and its alloy. It is very important the results of the ion release test in Ti6Al4V especially the vanadium release which can affect its cytotoxicity. A pargraph has been introduced according to the reviewer.
Introduce the abbreviations first.
The abbreviations have been introduced after abstract.
Ethical approval for animal study, was it taken?
The ethics committee of our university commented that the approval of the ethics committee is not necessary because they are not human samples, nor are they tested on live animals or animals that have been killed for this research. The ribs were purchased from a butcher's shop.
Do not start the sentence with Figure 1 and so on
This sentence has been changed according to the reviewer.
Clear and zoomed in, well copped photos of implants in cow ribs would be better, in figure 2, the 4 photos need to be highlighted in caption as well,
Figure 2 has been improved and a new legend figure incorporated explaining the process.
Figure 3- explain and label the figure to explain what it shows.
Done
Extractions for analysis were performed at 1, 3, 7, 7, 14 and 21 days following other similar studies [XXX], please correct XXX?
The references have been introduced in the text.
Figure 8, any clear fill photo available to put in with full margins?
Done. A new figure with full margins and more clear has been changed.
Add in the limitation and future recommendation for your study in clinical practicality based on your results.
The authors have introduced new paragraphs in the introduction with new references and a new section about the limitations of this contribution and the future studies
Reviewer 4 Report
Dear Callejas et al.,
The manuscript “Characterization of titanium particles released from bone-level titanium dental implants. Effect of the size particles on the ion release and cytotoxicity behaviour.” (materials-1715767) by Callejas et al. aims to study the effect of size on the release of titanium ions into the physiological environment, i.e. their chemical degradation and how it affects the levels of cytocompatibility with fibroblastic and osteoblastic cells. The topic is interesting, but I think this article should reconsider after proper changes in major revision for publication in Materials. Some of my specific comments are below:
- In the abstract section, the authors should add quantitative results rather than only qualitative results.
- Why does the present study focus on titanium dental implants? Why not other materials? It needs to be more discussed in the introduction section.
- Describe the novelty of the article made by the author? From the results of my evaluation, it seems that many similar published works adequately explain what you have raised in the current manuscript related to the metal particle of dental implants. If there is something others really new in this manuscript, please highlight it more clearly in the introduction section.
- The state of the art and the significance of the current study are not clearly present, the authors should highlight it more advanced in the introduction section.
- Since this manuscript evaluates metal dental implant, I would encourage and advise the authors to adopt some of the specific additional references related to metal based implant by MDPI in the introduction section as follow:
-
- Tresca Stress Simulation of Metal-on-Metal Total Hip Arthroplasty during Normal Walking Activity. Materials (Basel). 2021, 14, 7554. https://doi.org/10.3390/ma14247554
- The Effect of Bottom Profile Dimples on the Femoral Head on Wear in Metal-on-Metal Total Hip Arthroplasty. Journal of Functional Biomaterials. 2021, 12, 38. https://doi.org/10.3390/jfb12020038
- In the last paragraph of introduction section, I found not effective statement “In this study we want to study…”. Please revise it.
- In the materials and methods section, the authors should add one systematic figure to illustrate the workflow of experimental testing in the present study to make the reader more interested and easier to understand rather than only using dominant text to explain.
- What is the reason for using Vega plus commercial dental implant? Is there nothing other option? Or this commercial implant is the best in market? Ot it have several advantages, such as competitive price?
- In figure 1 the authors show a rendering figure of titanium dental implant used in this study, I am encourage the authors can add an engineering drawing and/or section view of its model for more clear understanding in engineering perspective on the same figure or separated figure.
- In figure 2, the authors are recommended to add scale to make the reader know the real comparison of the size of the image in the figure with the original so that it can provide a good understanding.
- It is so strange to have double Figure 5 in the present manuscript, please check it.
- Related to previous comments, having three separated figure with the relative same information is not worthy in scientific presentation om the journal. I encourage the authors to combine three of the figure into one, also giving the legend for different granulometry size.
- The author must provide a detailed specification and use condition more detail regarding all tools used in the research carried out so that the reader can estimate the accuracy and differences in the results that the authors describe due to the use of different tools in future studies.
- In the Results and discussion section, the authors are advised to compare the results they obtain with previous similar/identical studies if it is possible.
- In the last paragraph before conclusion section, the authors should add of one paragraph about the limitations of the presented review.
- The conclusion of the present manuscript is not solid. Further elaboration is needed.
- Further research needs to be explained in the conclusion section.
- In the whole of the manuscript, the authors sometimes made a paragraph only consisting of one or two sentences that made the explanation not clearly understood. The authors need to extend their explanation to become a more comprehensive paragraph. In one paragraph, it is recommended to consist of at least 3 sentences with 1 sentence as the main sentence and the other sentences as supporting sentences.
- I see some errors on English in some areas of the present manuscript. To improve the quality of English used in this manuscript and make sure English language, grammar, punctuation, spelling, and overall style are correct, further proofreading is needed. As an alternative, the authors can use the MDPI English proofreading service for this issue.
- The template of Materials, MDPI is not used correctly. The authors can download published manuscripts by Materials, MDPI, and compare them with the present author's manuscript to ensure typesetting is appropriate.
I am pleased to have been able to review the author's present manuscript. Hopefully, the author can revise the current manuscript as well as possible so that it becomes even better. Good luck for the author's work and effort.
Best regards,
The Reviewer
Author Response
REVIEWER 4
Dear Reviewer,
Thanks for taking the time to review our manuscript and suggest to us to improve our work by providing a lot more detail. We have done so, and we are now submitting a manuscript that not only addresses the points the you specifically raised but also many others that we have considered in order to deliver what we think is a much improved version of our work. This version includes more paragraphs, English grammar revisions in all main sections, new references. Thanks a lot. We are looking forward to your comments.
Sincerely,
Francisco-Javier Gil Mur
The manuscript “Characterization of titanium particles released from bone-level titanium dental implants. Effect of the size particles on the ion release and cytotoxicity behaviour.” (materials-1715767) by Callejas et al. aims to study the effect of size on the release of titanium ions into the physiological environment, i.e. their chemical degradation and how it affects the levels of cytocompatibility with fibroblastic and osteoblastic cells. The topic is interesting, but I think this article should reconsider after proper changes in major revision for publication in Materials. Some of my specific comments are below:
- In the abstract section, the authors should add quantitative results rather than only qualitative results.
Abstract has been changed according to the reviewer adding quantitative results and explaining the originality and the interest of the contribution.
- Why does the present study focus on titanium dental implants? Why not other materials? It needs to be more discussed in the introduction section.
Titanium is the most widely used (90-95% aprox) in comparison with other titanium alloys and zircona. This reason has been introduced in the text.In addition, two references have been added.
- Describe the novelty of the article made by the author? From the results of my evaluation, it seems that many similar published works adequately explain what you have raised in the current manuscript related to the metal particle of dental implants. If there is something others really new in this manuscript, please highlight it more clearly in the introduction section.
In agreement with the reviewer, the authors have added different paragraphs in the introduction. The relationship between particles released into the physiological environment and their relation to peri-implantitis disease is explained as well as the lack of characterisation of particles released by frictional forces in the placement of bone-level dental implants.
- The state of the art and the significance of the current study are not clearly present, the authors should highlight it more advanced in the introduction section.
This aspect has been added in the introduction with 7 new references.
- Since this manuscript evaluates metal dental implant, I would encourage and advise the authors to adopt some of the specific additional references related to metal based implant by MDPI in the introduction section as follow:
- Tresca Stress Simulation of Metal-on-Metal Total Hip Arthroplasty during Normal Walking Activity. Materials (Basel). 2021, 14, 7554. https://doi.org/10.3390/ma14247554
- The Effect of Bottom Profile Dimples on the Femoral Head on Wear in Metal-on-Metal Total Hip Arthroplasty. Journal of Functional Biomaterials. 2021, 12, 38. https://doi.org/10.3390/jfb12020038.
These paper has been referenced in the text
- In the last paragraph of introduction section, I found not effective statement “In this study we want to study…”. Please revise it.
The paragraph has been revised.
- In the materials and methods section, the authors should add one systematic figure to illustrate the workflow of experimental testing in the present study to make the reader more interested and easier to understand rather than only using dominant text to explain.
In agreement with the reviewer we have explained in more detail the materials and methods of the work. Hopefully this will be enough to avoid another figure, as another reviewer commented that the article had too many figures. Thank you very much for your understanding.
- What is the reason for using Vega plus commercial dental implant? Is there nothing other option? Or this commercial implant is the best in market? Ot it have several advantages, such as competitive price?
The Vega plus dental implant is a high quality implant and its characteristics have seemed to us to reflect the most common designs in bone level implants. It is a CE and FDA marked dental implant and is currently sold worldwide with a very low failure rate.
- In figure 1 the authors show a rendering figure of titanium dental implant used in this study, I am encourage the authors can add an engineering drawing and/or section view of its model for more clear understanding in engineering perspective on the same figure or separated figure.
We have tried to get the company to provide us with the plans of the implant but they are under protection and we have not been able to obtain the schematic. However, we have obtained the most important parameters of the implant that can influence the work. The roughness, Ra and Rz values as well as the procedure to obtain, the sandblasting material, the pressure, the diameter of the neck, the length, .... I hope that this data is enough, otherwise the authors will make a scheme if the reviewer wishes.
- In figure 2, the authors are recommended to add scale to make the reader know the real comparison of the size of the image in the figure with the original so that it can provide a good understanding.
According to the reviewer we have added more details about the ribs, length, homogeneity of bone density, placement of implants with neck diameter, placement distances between implants to avoid modifications of the bone characteristics among the most remarkable aspects. We think that with these measurements of the ribs, the diameter of the necks of the dental implants and the distances between implants we give the reader an idea of the dimensions, which we think will be easier than with a micrometer scale. However, if the reviewer thinks that it is not enough we would make the micrometric line.
- It is so strange to have double Figure 5 in the present manuscript, please check it.
The number of the figure repeated has been changed.
- Related to previous comments, having three separated figure with the relative same information is not worthy in scientific presentation om the journal. I encourage the authors to combine three of the figure into one, also giving the legend for different granulometry size.
Done.In the new version there is only one figure.
- The author must provide a detailed specification and use condition more detail regarding all tools used in the research carried out so that the reader can estimate the accuracy and differences in the results that the authors describe due to the use of different tools in future studies.
Done
- In the Results and discussion section, the authors are advised to compare the results they obtain with previous similar/identical studies if it is possible.
A paragraph has been added at the end of the discussion comparing the results as commented by the reviewer.
- In the last paragraph before conclusion section, the authors should add of one paragraph about the limitations of the presented review.
In agreement with the reviewer, the authors have added a discussion paragraph commenting on the differences found for titanium dental implants for one measurement, but highlighting the ion release study where the better performance of commercially pure titanium than Ti6Al4V alloy can be appreciated.
- The conclusion of the present manuscript is not solid. Further elaboration is needed.
The conclusions have been changed according to the reviewer. In this new version all sentences are based on the scientific results of the paper.
- Further research needs to be explained in the conclusion section.
- The authors have introduced new paragraphs in the introduction with new references and a new section about the limitations of this contribution and the future studies
- In the whole of the manuscript, the authors sometimes made a paragraph only consisting of one or two sentences that made the explanation not clearly understood. The authors need to extend their explanation to become a more comprehensive paragraph. In one paragraph, it is recommended to consist of at least 3 sentences with 1 sentence as the main sentence and the other sentences as supporting sentences.
The text according to the reviewer has been improved. Thanks for the comment.
- I see some errors on English in some areas of the present manuscript. To improve the quality of English used in this manuscript and make sure English language, grammar, punctuation, spelling, and overall style are correct, further proofreading is needed. As an alternative, the authors can use the MDPI English proofreading service for this issue.
Thank you for your information. We have improved the English with a native researcher.
- The template of Materials, MDPI is not used correctly. The authors can download published manuscripts by Materials, MDPI, and compare them with the present author's manuscript to ensure typesetting is appropriate.
The authors have changed the template according to the reviewer comment.
I am pleased to have been able to review the author's present manuscript. Hopefully, the author can revise the current manuscript as well as possible so that it becomes even better. Good luck for the author's work and effort.
Round 2
Reviewer 1 Report
- For the legend of Figure 7. "Ion release for the different sizes of c.p.Ti particles at different testing times. 353 (* means differences statistical significance with p<0,1)" Usually, The lowest p-value for statistical difference is 0.05.
- For the legend of Figure 8. "Cellular viability in cytotoxicity tests. a. Osteoblasts SAOS-2 cells. b. Fibro-368 blasts HFF-1. The results do not present statistical significance differences p<0.001." Please provide the data of the p value.
Author Response
Thank you very much for your comment. In accordance with the reviewer's comments, the authors have added the p-values for each of the cell types studied in Figure 8. In Figure 7 we have added the reviewer's comment and indicated the p-values for the 30 micrometre particles, which show the greatest differences with respect to the small particles. We have also entered the p-values.
Reviewer 2 Report
Dear authors,
Your reply and manuscript re-submission have answered to most of previously raised points.
Indeed, an extensive English writing editing is still necessary to improve the quality of the manuscript in order to be accepted.
Authors are also invited to re-write Introduction in a more concise and structured way.
In my opinion Limitations section should be included in the Discussion part.
Author Response
Thank you very much for your comments.
1. We have better structured the part of the introduction with the modification of the bibliographical citations.
2. The limitations of the study and future lines of research have been introduced in the Discussion according to the reviewer's criteria.
3. The English has been improved and errors in some words have been corrected.
Reviewer 4 Report
Dear Callejas et al.,
After carefully reading the author's revised manuscript entitled "Characterization of titanium particles released from bone-level titanium dental implants. Effect of the size particles on the ion release and cytotoxicity behaviour." (materials-1715767) by Callejas et al., The authors have been made significant improvements in the revised manuscript. Also, all of the issues in my review report have been addressed precisely.
With my pleasure, I recommend the manuscript should be accepted for publication on Materials.
Best regards,
The Reviewer
Author Response
Thank you very much for your comments.